# Mapping Food and Health Premises in Barcelona. An Approach to Logics of Distribution and Proximity of Essential Urban Services

**Carles Crosas and Eulàlia Gómez-Escoda \*** 

Barcelona Urbanism Laboratory, 08028 Barcelona, Spain; carles.crosas@upc.edu
\* Correspondence: eulalia.gomez@upc.edu

**Abstract:** The research analyzes the image of Barcelona and compares differences in quantity, variety and proximity of some essential services in diverse urban fragments. Focusing on food and health premises as critical universal services, series of maps provide overviews on the intensity of use to which each service is subjected, latent logics of their physical proximity and performance in regular urban fabrics due to the combination of activities and population distribution. The research uses a methodological approach and parameterization of the minimum daily urban mixture to highlight the uniqueness of the case of Barcelona, distinguished by the compactness of the urban fabric and the contiguity of activities, and to describe an extensive characterization of areas that from this perspective can be considered hyper-served or under-served. This investigation aims to contribute to the understanding of the necessity of the urban mixture and to provide clues about the distribution of services and activities.

**Keywords:** urban mixture; urban activities; commerce; essential services; Barcelona

## 1. Introduction

This article analyses the distribution of essential services in Barcelona with the aim of highlighting the value of minimum mixes of daily activities. The investigation is influenced by the anomalous operation of the city during the lockdown: on 14 March 2020, due to the Covid-19 pandemic, the Spanish government decreed a national lockdown in which only essential services were allowed to continue operating: food stores, pharmacies and health facilities were, during a period of eight weeks, some of the few blinds that were raised at ground level in the city. The new "stay at home" routine radically changed urban performance with an unseen reduction of mobility and a radical switch of the social network to the digital scene. It was a syncopated global phenomenon: within a few weeks, cities worldwide readapted their everyday life following unexpected patterns to reduce the pandemic's spread. The lockdown revealed the qualities and deficiencies in the distribution of essential services in relation to the density of the residential population, often masked by forced daily mobility, and transferred to neighborhoods a domestic role at the urban sphere.

According to mobile phone data [1], Barcelona switched from hosting 2,337,104 people on 13 March 2020 to 1,634,957 two days after. Once most of the workplaces had closed, the urban debate during the lockdown focused on households and streets: some features of the housing stock were questioned as the sole daily scene for the majority of citizens, in terms of size [2], as common working places [3] and emphasizing the value of the users in their wise adaptation [4] making the most of any outdoor contact spaces such as window lintels and balconies. In addition, the drastic reduction of 77% of vehicle movements [5] highlighted the unprecedented quality of air (both in terms of pollution and visibility) and an explosion of greenery in the city.

In the locked-down city [6–8], closeness became an appreciated asset because it ensured universal and safe supply: it minimized time away from home and facilitated access to basic products. In a new urban scenario characterized by empty streets, the layout of commerce became as much or even more important than in the active city itself, because it turned the "additional value" that commerce represents into "essential value" in the exceptional context.

Beyond the exceptional context, urban proximity has become an important concept in the urban agenda for a more livable city, with on-trend concepts such as the "15-min city" [9,10]. Proximity has been the key to many other paradigms over the history of urbanism. The concept focuses on setting the conditions to cope with all needs for everyday life in the closest areas to citizens' households. Proximity guarantees shorter trips and promotes non-motorized transportation, reducing pollution and congestion problems; it creates a more democratic urban space, diminishing the social differences caused by diverse access to transport; it makes all the facilities in the city equally available to all kinds of people, and ends up producing equal and socially sustainable travelling patterns [11].

## 2. Materials and Methods

The research deciphers the rationale for establishing minimum services all over Barcelona and discusses their relation to population and distances. The investigation produces thematic maps that relate some of the minimum services with population density and urban geography. All the figures are based on maps detailing urban subplots that represent built volumes inside a plot in the format of a shapefile geospatial vector data file (SHP) [12]. The following paragraphs describe the urban planning and design manuals and georeferenced databases that have been used to draw up some basic layers that describe the number of inhabitants, the population density and the minimum essential services that are considered for this research. Calculation methods and a first description of the base maps are described below.

### 2.1. Materials and Methods to Calculate Population

This research has only taken as a measure for calculations resident population figures, disregarding journeys and floating population data for two reasons. On the one hand, available data based on mobile phones only consider journeys at distances of a minimum of 10 km for Barcelona [13], so they do not include mobility between neighborhoods and districts of the city. This fact could distort the results (as does not considering them at all). On the other hand, in the event that the description of essentiality had been limited to exceptional situations such as lockdown, during the 2020 lockdown forced mobility was residual: according to the results of the Covid-19 Survey in Barcelona [14] it is estimated that more than 70% of the active population of the city telecommuted during the lockdown period. Additionally, around 30% of the population worked in services that were considered essential, including food, pharmacy, health, transportation, security and the media. This radical decrease in daily mobility would come to draw a kind of accidental simplification of the aforementioned "15-min city" [1,9,10].

Two different sources were used for the population. The Municipal Population Register records a total of 1,666,530 inhabitants in 2020 and geolocates the population disaggregating the information at block level [15]. For densities, the last version of the Statistical Yearbook of Catalonia disaggregates information at the Nomenclature of Territorial Units (between 150,000 and 800,000 average population size, NUTS3) level, through an SHP data file with referenced population count in a $62.5 \times 62.5$ m dynamic grid. The most recent publication sets an amount of 1,608,746 inhabitants in 2016 [16]. The difference in total number between both databases has been considered irrelevant for the purposes of the calculations of this research.

### 2.2. Methods to Calculate Distances and Proximity

In recent times, the disciplinary debate on how to reduce daily traffic has been reinforced. The aim is to design self-sufficient mixed neighborhoods where it should be possible to live, work and have

basic services without major journeys. The discussion on urban communities and the local and nearby scale of the city has been quite present in Barcelona, where some urban polices were addressed "to become a self-sufficient city of productive neighborhoods at human speed, inside a hyper-connected zero-emissions Metropolitan Area" [17].

Regarding proximity, some authors have paid special attention to the relevance of micro-journeys in everyday metropolitan performance. Short trips are considered those that take less than a 10-min walk, which represents in Barcelona 23% of the total mobility performed by citizens between 16 and 64 years old [18]. According to the same research based on Everyday Mobility Inquiry [19] data, proximity trips are 64.4% of total trips for shopping purposes, whereas in the total amount of daily trips, proximity trips represent only 22.9%. Other interesting references specifically analyzed walking behavior for shopping purposes in other metropolitan areas [20,21]. From the perspective of daily trips, mobility enquiries such as EMQ06 take as a representative a 5- and 10-min walk, according to the aforementioned references and other sources (open source platforms [22,23] can be equivalent to a range between 330 and 400 m (5 min) and 650 and 800 m (10 min).

This research takes the threshold of 400 m in the consideration of proximity distances due to the relationship that this measure establishes with the urban form of Barcelona. The grid plan designed by Ildefonso Cerdà in 1859, to modernize the city once the walls had been demolished, chose an inter-distance between blocks of 133 m, so that the sum of three blocks coincides with these 400 m. Years later, between 1930 and 1935, the Macià Plan designed by the modern architecture group GATCPAC would take the distance of three Cerdà blocks as the basic structure of its proposal for the city. Inherited from this are the bridges that cross Gran Via, west of Glòries Square, every 400 m and the piers that interrupt the coastline, also every 400 m. In the most contemporary context, the 3 × 3 block matrix is at the origin of the superblocks project [24,25].

Additionally, for this research the approach involved the calculation of linear distances rather than isochrones based on the shape and topography of the city. Although this method is considered sufficient to give a global vision of the scale of the city, the next phases of the research will refine the data in this sense to obtain results that are more adjusted to the scale of the plot. The count of activities in each plot is established through a 400 m radius measured from the centroid of each plot. In the calculations of proximity between services, plots destined for public facilities, industrial buildings and port facilities have been excluded.

### 2.3. Materials and Methods on Distribution of Services

Recent research has analyzed the general logics of trade. In Barcelona, the groundscape definition [26] was established, illustrating different patterns depending on the more local or more general character. The local–metropolitan duality has also guided some studies to identify the main features and location of central spaces in metropolitan Barcelona, based on a multiple combination of the factors of accessibility, connectivity, concentration of activities, mixture and identity, among others, that crystallizes at points of greatest intensity in the territory [27]. From this perspective, the distribution of commercial services is explained here only in part in relation to the resident population, whereas other factors have a great impact in the area such as the degree of tertiarization, the concentration of jobs and the notable effects of tourism.

The Study of Commercial Activities in Barcelona 2019 [28], published as an open database, is taken as the main tool to quantify and geolocate activities for this research. In regular circumstances, in 2019 it was recorded that a total of 55,824 active establishments on the ground floor are shops and services, which represent 90% of the total of activities (the rest, up to 61,558, are dedicated to other administrative, industrial and facilities uses). Of these, around 12.8% (7851) correspond to food suppliers and 18.1% (11,155) to restoration and hospitality. The high concentration of establishments along the streets gives strong character to the city and certain percentages illustrate the differences in comparison with other metropolises: in Barcelona 94.53% of shops and services are located on the street, 3.49% in municipal markets, and only 1.45% in shopping centers and 0.53% in passages and galleries.

*2.4. Considerations on Essentiality*

Undoubtedly influenced by the exceptional circumstances that changed the functioning of cities around the world during the spring of 2020, this research chooses from the urban services that can be considered basic (those that provide access to food, health, education, culture, transport, or that guarantee contact with the natural environment) only those that in the first phase of the most severe lockdown in Barcelona remained open to the public while the rest of the city withdrew towards the domestic space.

Among services at street level considered essential during the urban lockdown, this research discards those that correspond to activities whose use by the majority of the population is considered neither direct nor essential (gas stations, transportation and storage). Food suppliers, pharmacies and health facilities were used most intensely, as they are used daily to cover the basic needs of citizens. The total number of these services is 8494 and they represent 88% of the 9604 establishments officially considered essential services (that, in turn, represent only 25% of the aforementioned active storefronts at ground floor level). For the purposes of this research, they are described from two complementary perspectives: in terms of absolute and relative quantification, and in terms of territorial location (as shown in Figure 1).

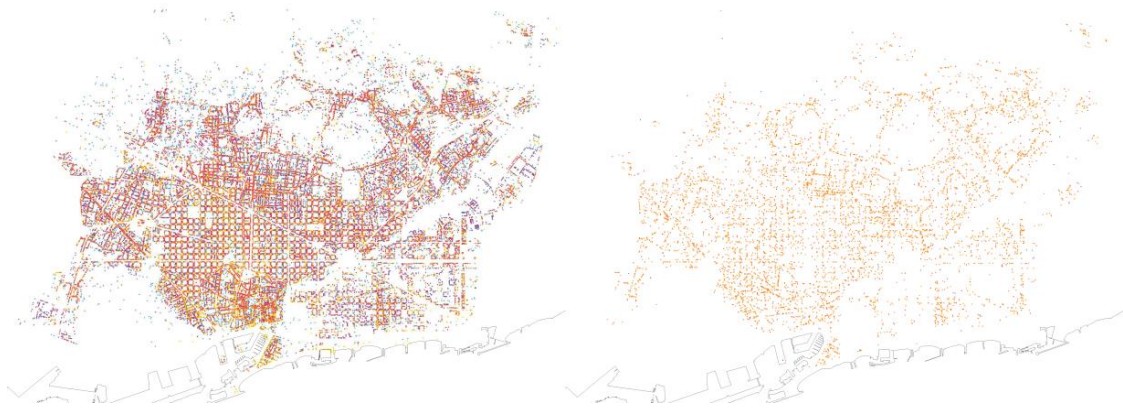

**Figure 1.** On the left, activities at street level in Barcelona (food suppliers in orange, retail in red, facilities in blue, hospitality in yellow and other services in purple). On the right, the essential services analyzed in this article (representing with the same color markets, supermarkets, grocery stores, bakeries, healthcare facilities and pharmacies).

Food supply systems constitute the main activity at ground floor level. Of the total establishments officially considered essential, 5928 (69%) correspond to food suppliers. This brings to an exceptional end the thesis that the urban form is closely related to food supply systems and that the way a city feeds its citizens constitutes a unique "foodprint" that provides information about its density, activity and equity [29]. The description of the Barcelona food system largely correlates with the explanation of the city and reveals how food suppliers "reflect location-responsive spatial qualities which have been developed and expressed over the long term" and respond to "time-tested design features that are argued to contribute to a food related sense of place" [30].

In this case, the uniqueness of Barcelona is represented by a double scale and a double public–private condition. First, the city has had a historic municipal commitment with citizens to guarantee access to fresh food, shown in the 38 buildings of municipal market halls which contain a total of 1415 stalls. Market halls are distributed in the territory in such a way that almost 40% of the inhabitants have a market at a distance of less than 400 meters [31]. Second, a constellation of scattered premises complement the market halls. They are composed of 2007 specialty grocery stores supplying fruit, fish and seafood, meat and pork or eggs and poultry and 2337 supermarkets in which each store

sells a varied mix of edibles, corresponding to the aforementioned categories for grocery stores plus processed and packaged foods and other cleaning and drugstore products.

Although they also provide food, bakeries deserve a special mention since their service role is controversial: traditionally a basic service, they have become fast-food places linked to tourism and low-cost urban exploitation as the city opened up to the world. For centuries, bread was the edible that guaranteed the survival of citizens living in poverty [32], and Barcelona City Council deployed different strategies to prioritize food safety over freedom of trade and began to buy flour to guarantee a minimum stock. Now numbered at 1562, bakeries are the only essential activity largely dedicated to tourists and office workers, preparing sandwiches and other fast meals to take away or eat in place, while continuing to fulfil their main function of selling bread and pastries to consume at home.

In terms of quantification and taking as a reference the 1,666,530 Barcelonans officially registered in the population census in 2020, the multiscalarity of the system also translates into a multiplicity of intensities in uses: each supermarket is used by an average of 713 inhabitants while market halls, whose exceptional character is based nowadays on their ability to bring together local centralities rather than exclusively supplying food, have an intensity of use more than 60 times greater, serving an average of 43,856 citizens. As a reference measure, the reader can take into account that of the ten districts of Barcelona, the least populated is Les Corts with 82,182 inhabitants (and one single market hall) and the most populated, the Eixample, with 266,754 (and five market halls). They both are above the average mentioned in the text, as is Sants (with 182,184 inhabitants and three market halls). Conversely, Nou Barris, with seven market halls in its demarcation and 168,327 inhabitants or Gràcia (121,593 inhabitants, four markets) are significantly below this median.

Each grocery store (in a theoretical exercise that does not take into account what type of product each specializes in and, therefore, ignores the necessary complementariness between establishments of this category) serves 830 inhabitants (a figure higher than that of supermarkets, which is significant if the smaller scale of these establishments is taken into account). Each bakery serves 1066 Barcelonans (to which must be added, in unconfined city conditions, the floating population that these establishments potentially feed every day).

Although food is the most widespread daily service, health services and pharmacies have a specific care role for a majority of the population and are places of reference for the most vulnerable citizens. Pharmacies in urban areas are normally regulated in number so that there must be a maximum of one for every 4000 inhabitants and at a maximum proximity of 250 m with respect to any other [33]. Given these rules, Barcelona would have 416 pharmacies according to its population. However, the census of establishments updated for that same year provides ratios more than 2.6 times higher, giving a total of 1086 pharmacies in the city, that is, one for every 1534 inhabitants. In the same way, distances between establishments are shorter in many areas of the city, as the regulations only apply to new openings.

Finally, the analysis of healthcare facilities deserves separate consideration, since the distribution they follow corresponds to specific criteria in accordance with their exceptional nature in terms of standard everyday life. In Barcelona, they represent one of the types of public community facilities recognized in the current General Metropolitan Plan (1976), together with those for educational services, cultural and religious activities, leisure and sports, provisioning and supply, and technical-administrative and security. The same planning document classifies them according to their sphere of influence into local facilities or facilities of supralocal or metropolitan interest. In the research, only two types of healthcare facilities have been considered as they offer a universal medical service: metropolitan-scale services that exceed any consideration of proximity (16 hospitals and 18 clinics and private medical centers), and the network of 53 health centers and primary care facilities that provide service in a neighborhood influence (which would mean, in an ideal balanced distribution, 31,443 Barcelonans per center).

## 2.5. Methods to Relate Density of Population and Distribution of Activities

The research relates the spatial distribution of activities and population densities. In a first step, a locus-approach strategy is applied. Using Voronoi diagrams, the query space (considering all the municipality but obviating facilities and open spaces) is partitioned into regions on which the distance to a specific activity is the same.

Each activity is considered a simple point (p, located in the center of the plot) and its corresponding Voronoi cell consists of every point in the Euclidean plane whose distance to p is less than or equal to its distance to any other p. Cells comprising the final map are obtained from the intersection of half-spaces and are delimited by line segments that represent all the points in the plane that are equidistant to the two nearest sites. Vertices in the maps are those points equidistant to three (or more) p activities.

## 3. Results

The research starts from a series of descriptive thematic maps and questions them to progressively reach some conclusions. Some previous considerations (both in terms of the general shape of the city and the analysis methodology used) are taken into account and can be visually compared in Figure 2.

- Given that the study is carried out strictly within the scope of the Barcelona municipal area, the approach does not consider the real metropolitan city, which distorts the results in the fringes close to the boundaries with the municipalities of L'Hospitalet de Llobregat, Esplugues de Llobregat and Cornellà de Llobregat in the southwest, and Sant Adrià del Besòs and Santa Coloma de Gramenet in the northeast. This methodological restriction, which is common in so many studies (due to the origin and nature of the available data), is not very distorting in this case given that in certain periods the municipality was a lockdown unit.
- The shape of the geography determines the rest of the limits of the analysis perimeter: the sea with the port at the front and Collserola mountain slopes as the back limit of the upper neighborhoods. Furthermore, some geographical singularities lead to service gaps in the drawings when they coincide with the paradigmatic hills of the city: the prominent Montjuïc and the Three Hills.
- Within the boundaries of the municipality, a number of large gaps or "empty" spaces are remarkable for the analysis results. For instance, unoccupied hills such as Montjuïc and the Three Hills, the historically extensive parks (Ciutadella) together with the large future public spaces (Sagrera and Glòries) and many other minor equipped areas.

The distribution of ground floor activities follows two logics. First, each type of public facility (markets and health services) follows a particular strategy that is explained below. Second, smaller-scale private retail services in four of the studied categories (supermarkets, grocery stores, bakeries and pharmacies) have some common distinctive features. Without taking into account the surface area, the services they offer, or the scope of their activity, the distribution of the units allows some patterns to be determined that in most cases correspond to the underlying street layout.

At a uniform scale view, leaving aside the aforementioned large gaps (topography, public spaces and facilities), there are common unserved places in all categories, where, although the surface area varies, gaps are drawn due to the absence (or fewer units) of activity: large facilities, transportation hubs, industrial, productive or use transformation areas, tertiary and office hubs, commercial hubs, former "new downtowns" [34] and upper-middle class neighborhoods.

Apart from these notable gaps, the distribution of essential activities due to the quantity of services and the urban compactness on which they are located is drawn to outline the shape of the city. The analysis detects some differences to characterize each of the layers as shown in Figure 3 (where from left to right and top to bottom the layers are displayed as follows: market halls in purple; bakeries in brown; supermarkets in yellow; grocery stores in orange; pharmacies in red; and healthcare facilities in blue) that can be described according to the relationship they establish with some of the other uses and the peculiarities in the occupation of the urban ground floors. Each layer is distributed following

the general rules described below, while Table A1 in the Appendix A of this document gives a more detailed explanation with more precise place names for knowledgeable readers of the city.

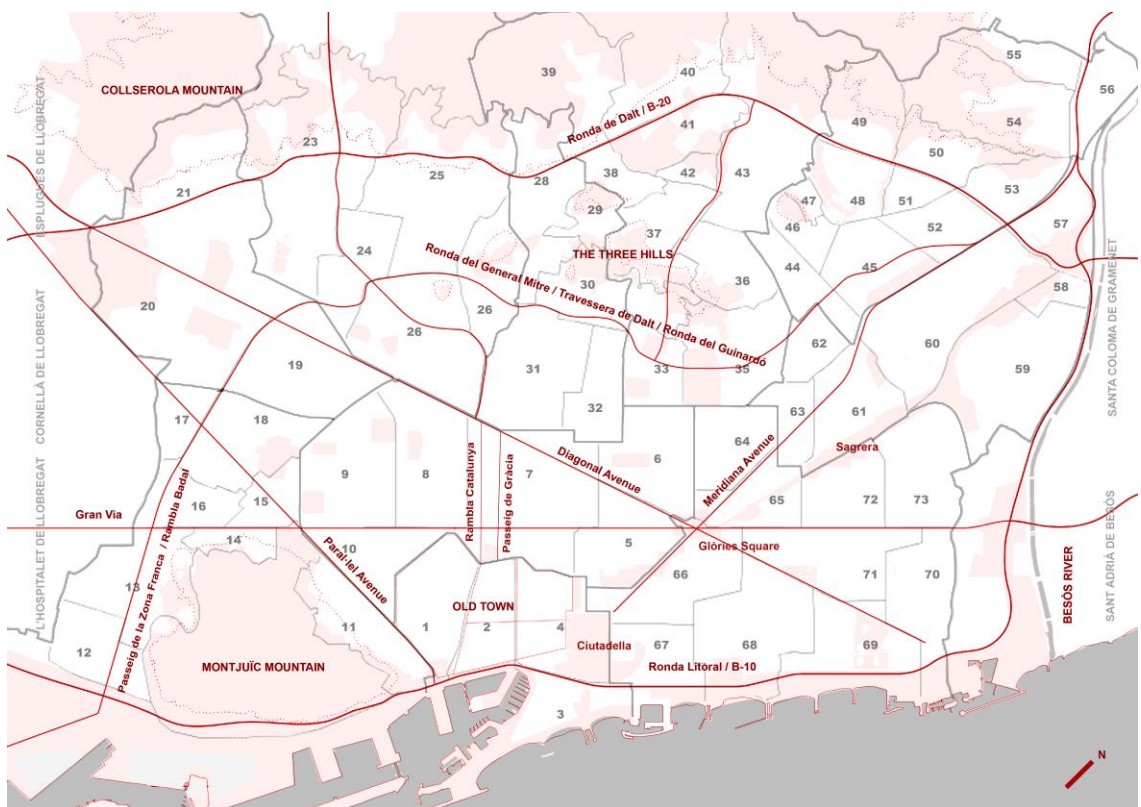

**Figure 2.** Map guide of the toponymy of Barcelona. Geography and main avenues are highlighted and named in red; the 73 neighborhoods composing the 10 districts are numbered as follows: in Ciutat Vella neighborhood: 1. Raval; 2. Gòtic; 3. Barceloneta; and 4. Sant Pere, Santa Caterina i la Ribera. In Eixample: 5. Fort Pienc; 6. Sagrada Família; 7. Dreta de l'Eixample; 8. Antiga Esquerra de l'Eixample; 9. Nova Esquerra de l'Eixample; and 10. Sant Antoni. In Sants-Montjuïc: 11. Poble Sec; 12. Marina del Prat Vermell; 13. Marina de Port; 14. Font de la Guatlla; 15. Hostafrancs; 16. Bordeta; 17. Sants-Badal; and 18. Sants. In Les Corts: 19. les Corts; 20. Maternitat i Sant Ramon; and 21. Pedralbes. In Sarrià-Sant Gervasi: 22. Vallvidrera, el Tibidabo i les Planes; 23. Sarrià; 24. Tres Torres; 25. Sant Gervasi-Bonanova; 26. Sant Gervasi-Galvany; and 27. Putxet i Farró. In Gràcia: 28. Vallcarca i els Penitents; 29. el Coll; 30. la Salut; 31. Vila de Gràcia; and 32. Camp d'en Grassot i Gràcia Nova. In Horta-Guinardó: 33. Baix Guinardó; 34. Can Baró; 35. Guinardó; 36. Font d'en Fargues; 37. Carmel; 38. Teixonera; 39. Sant Genís dels Agudells; 40. Montbau; 41. Vall d'Hebron; 42. Clota; and 43. Horta. In Nou Barris: 44. Vilapicina i la Torre Llobeta; 45. Porta; 46. Turó de la Peira; 47. Can Peguera; 48. Guineueta; 49. Canyelles; 50. Roquetes; 51. Verdun; 52. Prosperitat; 53. Trinitat Nova; 54. Torre Baró; 55. Ciutat Meridiana; and 56. Vallbona. In Sant Andreu: 57. Trinitat Vella; 58. Baró de Viver; 59. Bon Pastor; 60. Sant Andreu; 61. Sagrera; 62. Congrés i els Indians; and 63. Navas. And finally, in Sant Martí: 64. Camp de l'Arpa del Clot; 65. Clot; 66. Parc i la Llacuna del Poblenou; 67. Vila Olímpica del Poblenou; 68. Poblenou; 69. Diagonal Mar i el Front Marítim del Poblenou; 70. Besòs i Maresme; 71. Provençals del Poblenou; 72. Sant Martí de Provençals; and 73. la Verneda i la Pau.

- Even without planning traces, the location of market halls followed a logic that since the nineteenth century considered the compactness of the fabric and the density of the population, so that the service radii determined the relative positions [35]. Market halls are located at a minimum distance of between 492 and 1663 m from any other nearest market. However, the most significant data regarding their location concerns the distribution of the rest of the food distribution services

(grocery stores, supermarkets and bakeries): more than 50% of these stores are within a maximum of five minutes from at least one of the markets, 38% at a distance of ten minutes, 9.5% at fifteen minutes and only 1% of food suppliers serve areas even further away from market halls.

- Supermarkets are more concentrated in Raval, Barceloneta and Poble Sec, also along some main axes and avenues.

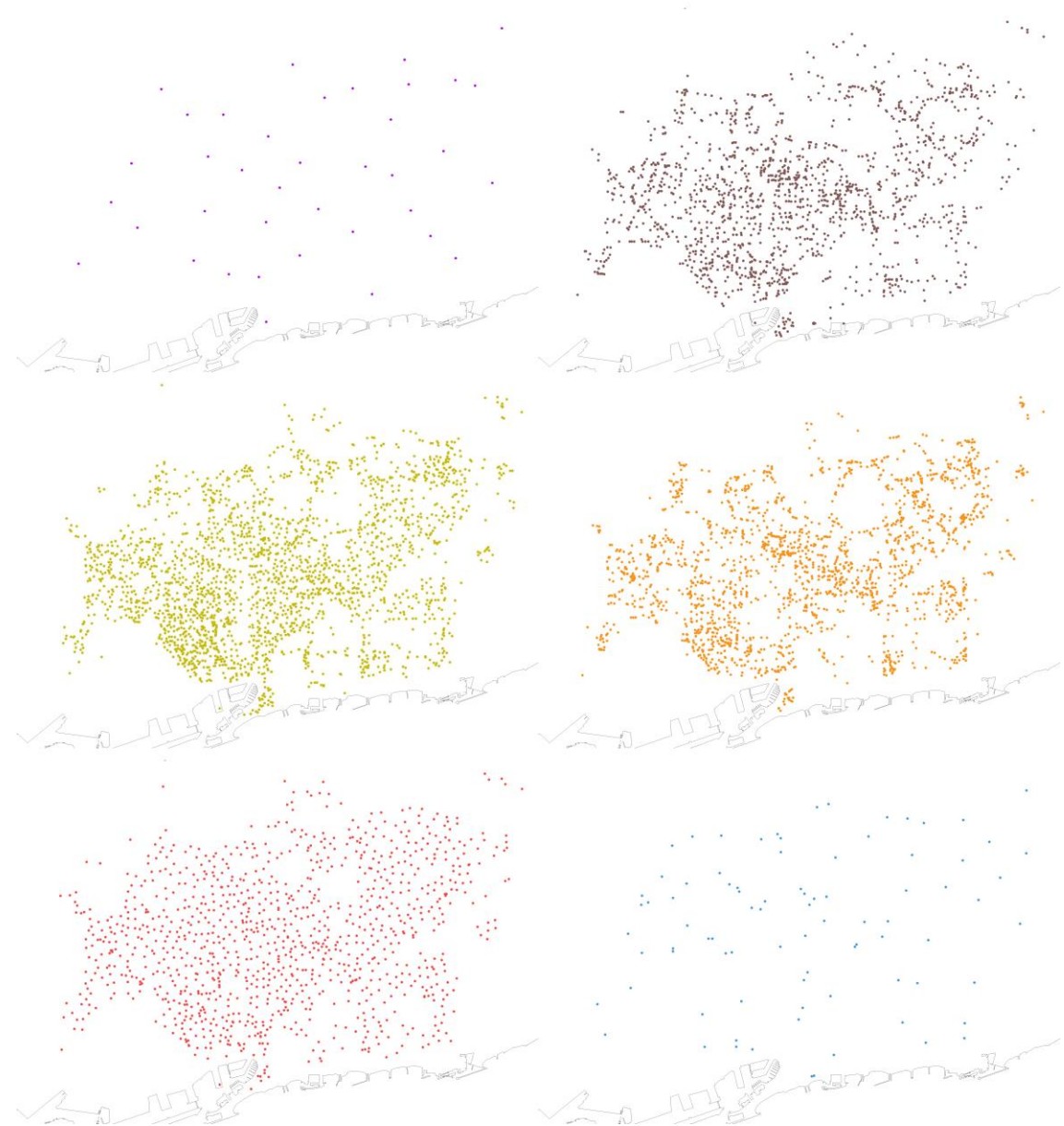

**Figure 3.** Distribution of essential services by categories (from left to right and top to bottom: market halls, bakeries, supermarkets, grocery stores, pharmacies and healthcare facilities). Table A1 in the Appendix A of this document gives a detailed explanation with precise place names for knowledgeable readers of the city.

- The nuclei of the old municipalities (Sants, Sarrià, Sant Andreu, Poblenou, Gràcia) and some streets emerge in the view of the footprint drawn by grocery stores.
- The concentration of bakeries follows the enclaves with the largest number of grocery stores, despite the fact that in this case they are less numerous and their image on the map of the city is less intense in repetition.

- Healthcare facilities are mainly concentrated on both sides of the Diagonal. There is a particularly intense cluster next to the Ronda del General Mitre.
- The pharmacies distribution diagram is the most isotropic of all, due to the equidistribution logic. Their territorial implantation pays more attention to the distances between the elements than to the specific shape of the urban fabric. In this case, it is interesting to consider their distribution related to health facilities locations: 41% of pharmacies are within a maximum of five minutes away from at least one health service, 52% at a distance of ten minutes and only around 6% at fifteen minutes or even further away. These numbers would give, in a hypothetical formula of perfect balance between the location of pharmacies and primary care centers, a distribution of 20 pharmacies in the immediate 10 min from each health center.

### 3.1. Mapping the Essential

Once the essential services have been analyzed in terms of distribution, the research focuses on comparing the differences in quantity, variety and proximity that they present in diverse urban fragments. To this end, a series of cartographies based on linear and Euclidean approximations are prepared. These should not be conceived as real walking distance indicators [36,37] as they obviate many of the factors involved in closer considerations (topography, accessibility, single act of purchase, subjectivity and security) that allow an evaluation of latent logics of territorial distribution for each service.

Three series of maps assess three complementary views on the topic: the intensity of use to which each service is subjected, the physical proximity between services and the alterations in regular urban fabrics due to different balances between activities through the specific case of the Eixample. All maps are elaborated under a specific methodology so the resulting graphic materials show a wide range of views on the topic. The formulation and partial results are described in the following subsections.

### 3.1.1. Density of Services and Intensity of Use

The first series of maps shows the spatial distribution of activities related to population densities. According to the method described in the previous section, a density of inhabitants is assigned to each Voronoi cell to obtain a theoretical model of allocation of the population to each service point. Figure 4 shows the resulting map of division of space into polygons and subsequent allocation of potential users to the tiles. The results for each of the six types of activity are divided into quintiles, and extreme values are taken for a new analysis. This gives two groups of activities designated as "over-assigned" (when the population served is in the top 20%) or "under-assigned" (in the bottom 20%).

The figures can be compared with the average allocations for each type of service in relation to the total population of Barcelona mentioned in Section 2: there are 702 people per supermarket, 804 per specialized grocery store, 1048 per bakery and 1507 per pharmacy.

Considering the over-assigned activities, the minimum number of inhabitants that set the maximum quintile ranges from 1001 people per supermarket, 1221 people per specialized grocery store, 1582 people per bakery, or 2035 people per pharmacy. In the assumption that the average service figure in the city ensures balance, the smaller the difference with the aforementioned average, the better balanced the service can be considered. In this way, pharmacies set their threshold for the maximum demand only 35% above the average, while supermarkets set it at 42%, and bakeries and specialized grocery stores, slightly above 50%.

In cases of under-assigned activities, the ranges are up to 318 people per supermarket, 261 people per specialized grocery store, 357 people per bakery or 882 people per pharmacy. This indicates that pharmacies are the most constant of the four services analyzed, since this quintile of minimum service remains at 58% of the average population supplied. At the other extreme are the specialized grocery stores, in which the figure drops to 32%, and bakeries, to 34%. In an almost medium position are supermarkets. In the quintile that serves the least amount of population, the figures for supermarkets stand at 45% of the average population served in the city.

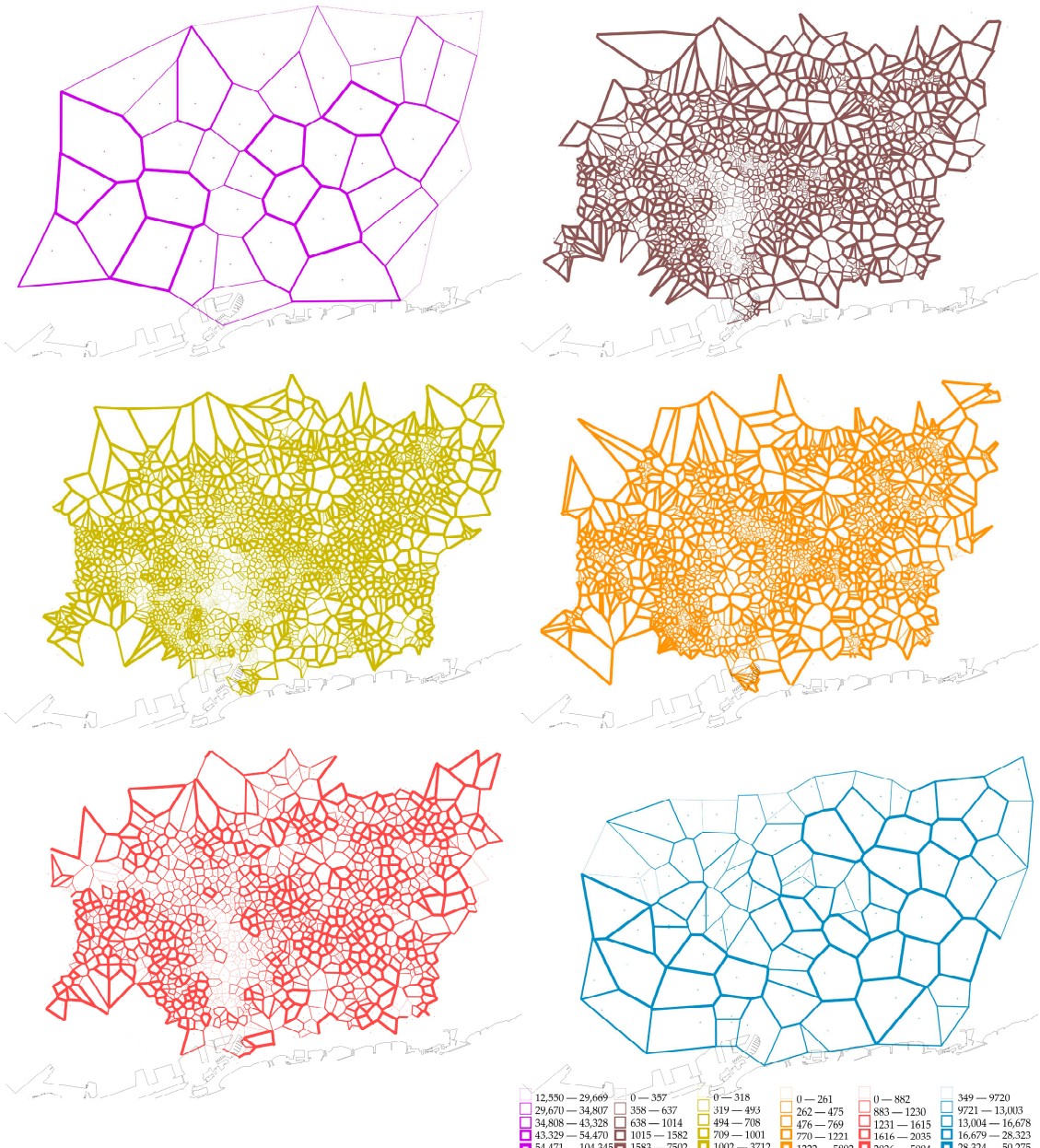

**Figure 4.** Voronoi diagrams of essential services by categories (from left to right and top to bottom: market halls, bakeries, supermarkets, grocery stores, pharmacies and healthcare facilities). Representation by quintiles with different line thicknesses (the thicker the line, the higher the population density assigned to a given service unit).

Figure 5 shows the territorialization by layers, representing on the urban fabric the cells that in Figure 4 corresponded to the highest quintile. This image allows the establishment of distinctions between activities that have a higher number of assigned inhabitants and are considered to have more potential to be in critical condition than those that have a lower density of assigned inhabitants:

- Given that, as explained above, market halls follow a fairly balanced territorial distribution logic, it is expected that nuances appear when looking from the perspective of how much population they serve, since they then account for the heterogeneity of the fabrics in which they are inserted.
- In the case of supermarkets, those that are supposed to serve a greater number of people are scattered throughout the city. However, some fragments of significant axes are highlighted in the

drawing. In addition, a series of other sectors show a high density of discontinuous tiles all over the city.

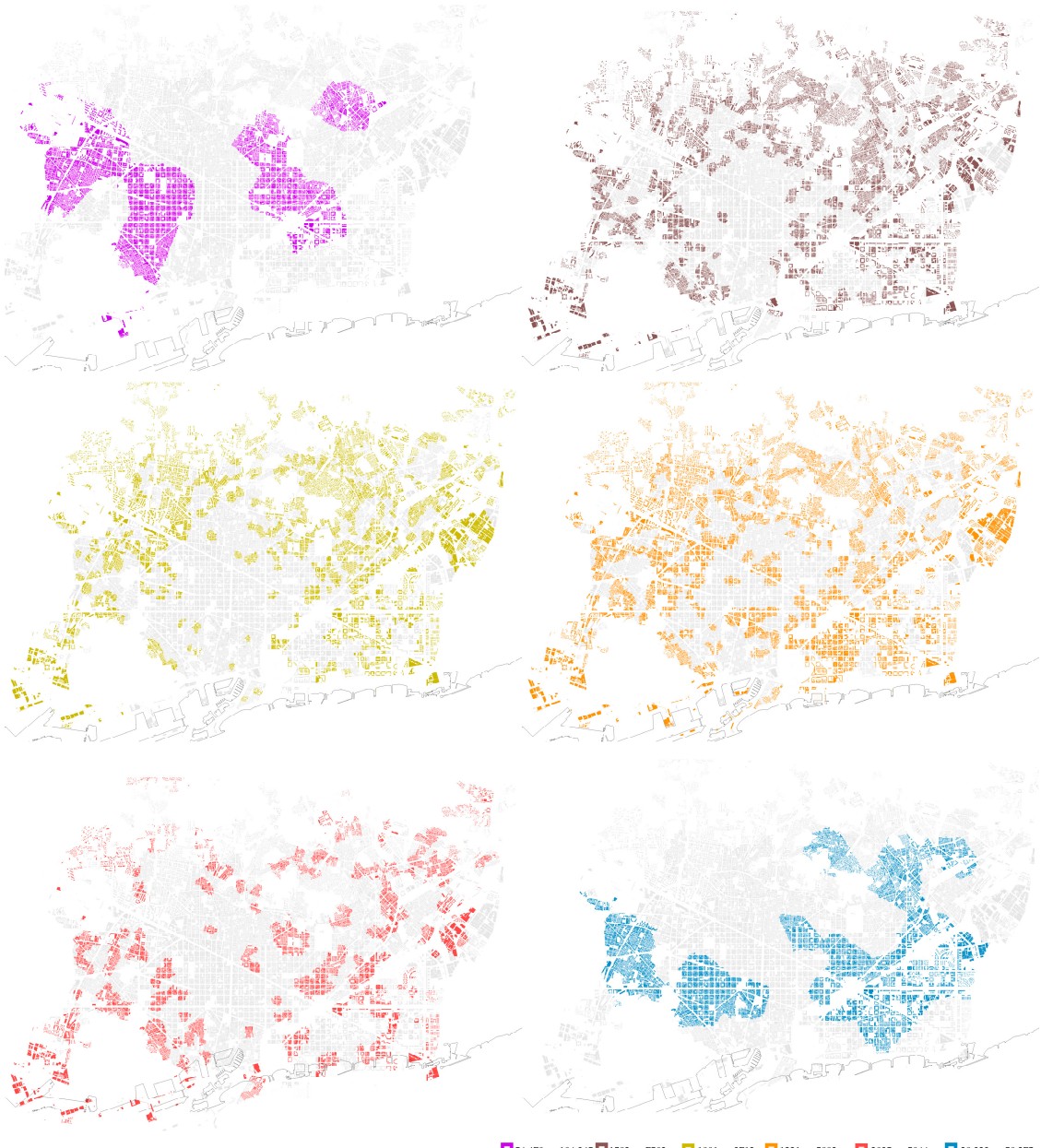

54,470 — 104,345  1582 — 7502   1001 — 3712   1221 — 5892   2035 — 5044   28,323 — 50,275

**Figure 5.** Places with the highest population allocation for each category of essential services (from left to right and top to bottom: market halls, bakeries, supermarkets, grocery stores, pharmacies and healthcare facilities). Representation of the upper quintile of Figure 4 in relation to the urban fabric. Table A1 in the Appendix A of this document gives a detailed explanation with precise place names for knowledgeable readers of the city.

- The layer of grocery stores is difficult to describe since several different types of specialty stores are considered under the same category. Broadly speaking, there is a greater concentration of areas in which the stores are assigned to a greater number of people north of the Diagonal. The central districts of the city (the entire Eixample and Ciutat Vella) have very few places where the Voronoi tiles indicate services with an over-assigned population.

- Bakeries reproduce the footprint of grocery store distribution with one exception: in the central areas there are no outstanding quintiles, since the number of establishments (that in a large number prepare breakfasts and take-away food for tourists and office workers) is greater.
- Healthcare facility premises that are over-assigned define a relatively continuous mosaic distributed around three large areas.
- The largest number of inhabitants corresponds to the most requested health centers. In this case, it is convenient to highlight two areas in which there is no over-allocation in health centers but there is in pharmacies.

### 3.1.2. Topography and Physical Proximity between Services

Figure 6 focuses on rectilinear distance to services, considering radii of 400 m from the geometric center of each one of the plots containing an activity. With very few exceptions, images show a 100% served city that can be explained by the short distances with which the services are distributed. Seen under the lens of the minimum distance, supermarkets and pharmacies in Barcelona share the same territorial distribution criteria: they are, on average, less than 100 m in a straight line from any plot in the city. Bakeries or specialized food stores reach a minimum average distance slightly higher than 120 m.

The proximity between services is such and the proportion of areas that are not served is so lateral that, once the guaranteed minimum service has been assumed, the discussion focuses on determining the contrasts between the amount of activities accessible from each individual plot. The quantification of the number of units of the same category that each plot has within the same 400 m radius provides a reference to understand further partial data: excluding facilities, industrial buildings and infrastructure services, each plot in the city has, on average in a 400 m radius, 24.86 supermarkets and almost the same amount of grocery stores (23.44), 17.05 bakeries and 10.8 pharmacies. This means that with the lens of urban daily services, all the plots in the city have 75 basic activities within five minutes' walking distance.

Given the logics of territorial implementation of each use, the determination of extreme values (top and bottom 20%) reveals plots that have a large number of services around them and, at the other end, plots with few daily activities in their vicinity. Looking separately at each of the four activities and dividing the results into quintiles shows what could be called hyper-served plots (when the number of activities accessible from a given plot is in the top 20% citywide) and underserved plots (in the bottom 20%). It must be stressed that this research looks at six essential urban services. Therefore, a number of many other activities that complement them are not included, so the qualifiers of hyper-service or under-service are relative to this exceptional still urban photography.

In hyper-served areas, plots have up to 92 grocery stores within 400 m, 79 supermarkets, 61 bakeries and 27 pharmacies. Compared to the average supply figures, this means values between 2.5 (for pharmacies) and 3.9 times higher (for supermarkets). Plots in under-served areas are not literally without any service around (approximately 50 hectares, 400 m radii) but they belong to the lowest quintile in which the numbers of establishments per category are less than: five bakeries, seven pharmacies, nine grocery stores, or eleven supermarkets. The range of differences in relation to the average is lower than in the case of hyper-served areas: pharmacies are only 64% below the aforementioned average, while bakeries decrease more than two thirds, up to 29%. Among them, grocery stores in these weaker areas are 38% below the mentioned average, and supermarkets, 44%.

A parallel reading of the 400 m radius series offers complementary information to the previous perspective. The higher the intensity of the color, the better served the areas in each category, whereas the faintest hatches indicate areas with minor service. At a glance and from this perspective, it is reconfirmed that a very substantial part of Barcelona has a very good level of essential services and that there are no broad differences among the mapping of the four categories (except markets and healthcare facilities). Actually, the maximum and medium service quintiles (darker shades) extend in

all cases from the old town upward, reaching in many cases urban fragments above the Ronda del Mig-Ronda del Guinardó.

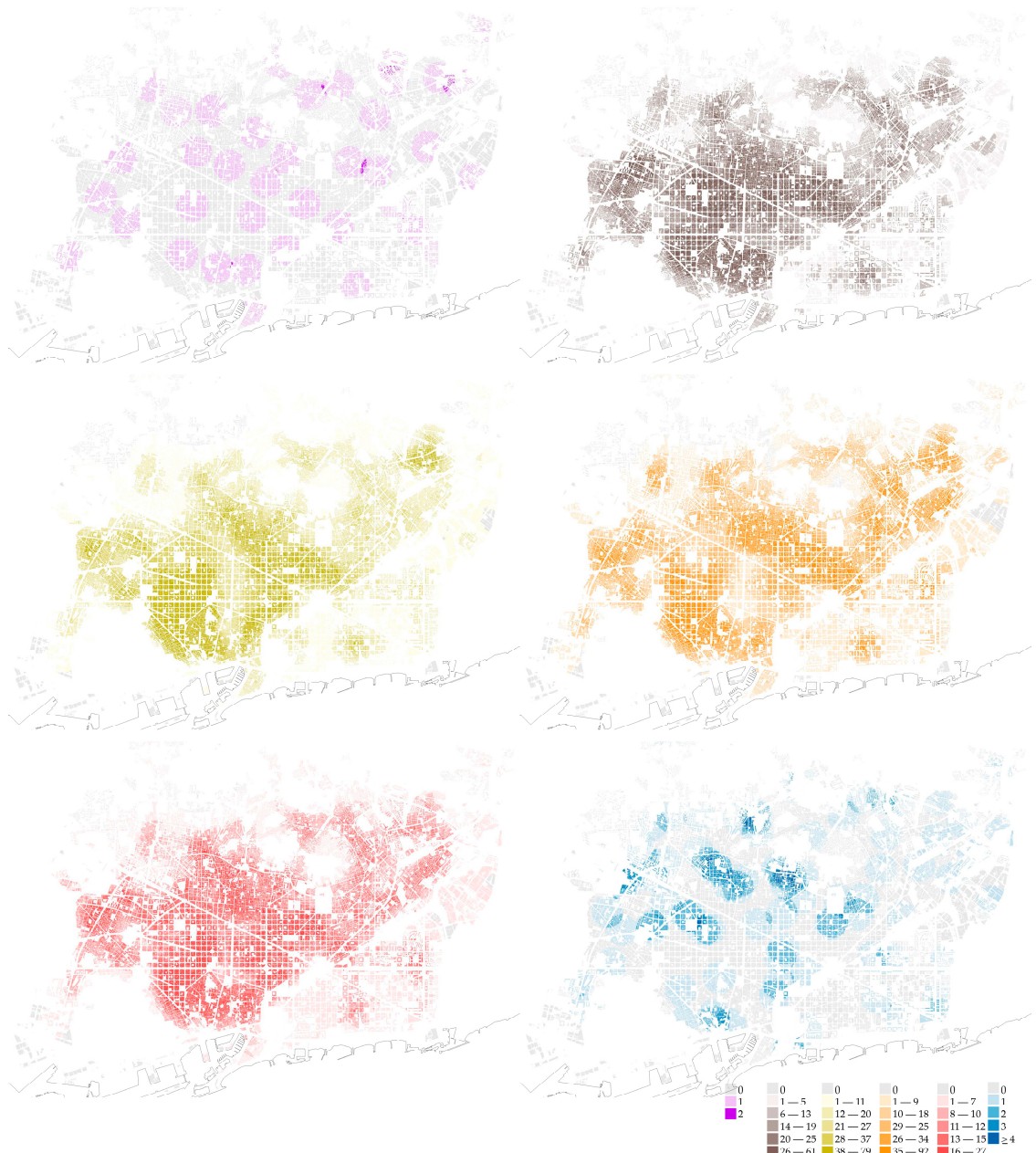

**Figure 6.** Urban fabric according to the proximity between essential services by category (from left to right and top to bottom: market halls, bakeries, supermarkets, grocery stores, pharmacies and healthcare facilities). Proximity between service units is represented by quintiles (the darker the hatch, the higher the concentration). Table A1 in the Appendix A of this document gives a detailed explanation with precise place names for knowledgeable readers of the city.

A quick overall comparison shows the most outstanding characterization of the outcome of the six maps. The most clear-cut cases are the two regarding facilities:

- Market halls service clearly illustrates their isolated condition and quite regular distribution that creates only some tiny overlaps either in the very central area or in the most peripheral.
- Healthcare facilities service intensifies along some axes.

Regarding pharmacies and bakeries, a relatively uniform intensity is shown in a wide central area that blurs towards the boundaries of the city. Avinguda Meridiana and most specifically the parallel railway tracks, from Ciutadella Park to Sagrera northwards, appear as an important inflection that signals historical limits between the older and the newer city, the most residential areas and the former industrial districts. Supermarkets and groceries service maps offer similar diagrams exaggerating two very singular situations: the Passeig de Gràcia area as a blank within dense hatches, and Carrer Marià Aguiló and Rambla del Poblenou as a pair of axes depicting the reverse situation, as hyper-served areas within the underserved tissues of Poblenou-22@.

The scarcity of essential services is summarized in a citywide overview of the underserved areas during the lockdown. It does not literally correspond to areas "without services" but to areas with the lowest number of services around (lowest quintile) in comparison with the whole city (see the aforementioned indexes quantification per category). The exercise of overlapping the four categories results in a map that reveals potential needs on the base of the absence of one of the services in relation to the almost total presence that the four of them have in the municipal built fabric, shown in Figure 7.

The overlapped view is described together with the topographical base of Barcelona in the background in order to provide some clues from this perspective. Actually, the cross-analysis illustrates once more the importance of the geographical support to determine the form and distribution of the activities over the city. It could be guessed that the more rugged the territory, the less served it is.

Identically, a similar basic principle could be taken regarding the logics of center–periphery. As Barcelona was founded in the middle of a vast plain, it is not strange that the level of services decreases towards the edges of the city because of two factors coinciding: central and flat areas versus peripheral and sloped zones. Nevertheless, looking at the central flat area, outstanding exceptions are mentioned below, while Table A2 in the Appendix A of this document gives a more detailed explanation with more precise place names for knowledgeable readers of the city:

- Areas previously pointed out for their past or current industrial use (or proximity to such uses) can be listed. In some cases, the overlap with the topography is representative of the tradition of the industrial buildings occupying the least healthy and humid areas, as was the case of the Besòs delta that was subsequently completely urbanized, but still recognizable in this map. Their current morphological configuration (block size, street layout) exemplifies different phases of transformation. Despite the fact that many activities can be normally found in their streetscape, the quantity of essential services is still quite low. The 22@ District fabric clearly expresses the underserved condition in opposition to the old Poblenou, where the intensity of greys (lack of service of one to four categories) emphasizes internal variations (Table A2).

- The main example of breaking the basic center–periphery principles is the Passeig de Gràcia strip. Around the central backbone of the Eixample, which is considered the epicentre of the city, there are paradoxically a number of underserved blocks. This question, already discussed in the previous sections, enriches the result of the research and underlines the effects of tertiarization and "globalization" of the commercial network in this area.

- Due to its position on the seafront (an edge from the methodological perspective), but also its own functional configuration, almost the entire waterfront is identified as an underserved area. This situation is strong in the renovated areas from La Vila Olímpica to Diagonal Mar, and has a lower presence in the traditional neighborhood of La Barceloneta, where only the plots on the edge share a similar condition.

- Finally, the underserved condition in the flat central area can be identified in some of the plain boundaries (for instance, around Montjuïc) and in some stretches alongside the infrastructural axis, such as Meridiana, Gran Via and the railway tracks at Sagrera.

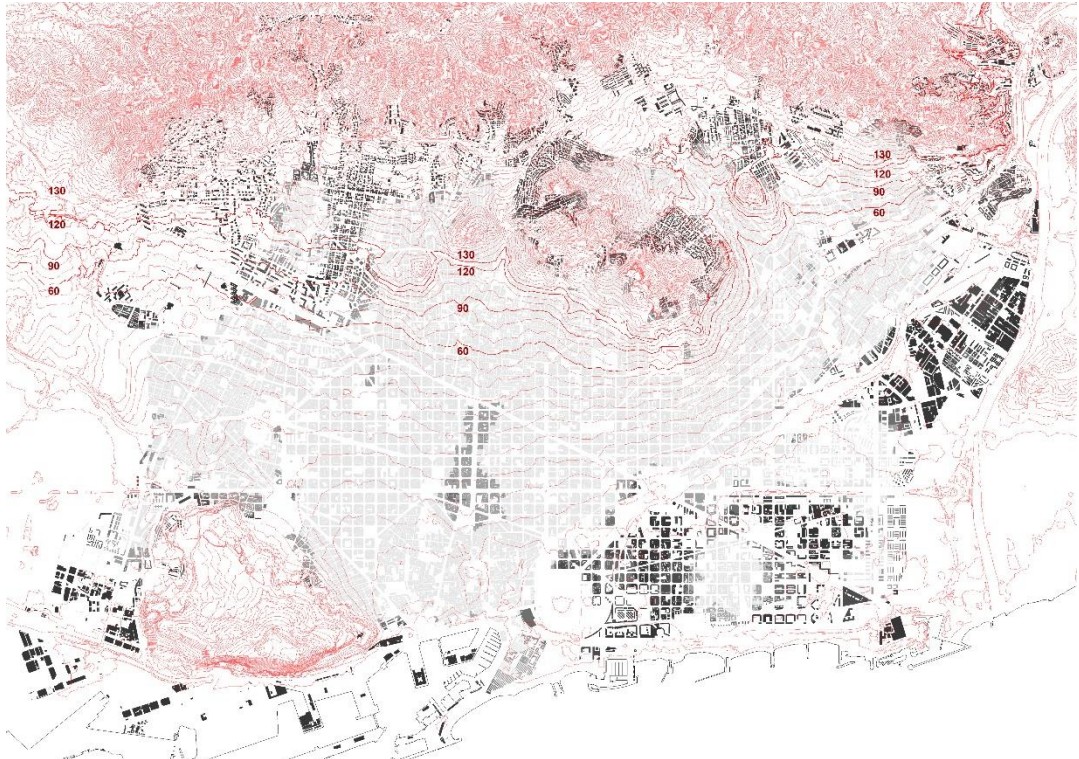

**Figure 7.** Areas with the fewest number of essential services (excluding facilities). Overlap of the lowest quintile of each of the categories resulting from Figure 6. Three categories are represented here with three different intensities: in black, areas in which the minimum intensity coincides in four of the services; in dark gray, in three of the services; in light gray, in two of them. The topography is superimposed in red and the urban background fabric in light gray. Table A2 in the Appendix A of this document gives a detailed explanation with precise place names for knowledgeable readers of the city.

Despite the many exceptions, the flat urban zone coincides with the well-served areas, whereas the highest urban fabrics next to the urban boundaries have in common their underserved condition. In between, the research shows a reference limit over which there is room for improvement due to the limited concentration of essential services.

A contour line at an altitude 60 m above sea level is a clear limit in the southern area—west Diagonal, whereas in the central area the limit references rise to lines of 120–130 and 90 m above sea level around the Guinardó district.

In the Three Hills area there are some important nuances among neighborhoods around the hills. While La Font d'en Fargas, La Teixonera, el Coll, Penitents and Vallcarca are areas with little service, the dense structure of El Carmel distinguishes it from this category.

Finally, in the northern part of the city, the critical situation is observed in three very different neighborhoods from a morphological point of view. The low proportion of services is common in Ciutat Meridiana, Torre Baró, Trinitat Nova and Trinitat Vella with quite different population densities but a similar proximity to the city borders.

### 3.1.3. Regularity and Intensity of Mixture

Due to its extension and homogeneity, the Eixample district is a paradigm observed repeatedly in urban research on Barcelona. In this case, the choice of the district as the scope for a detailed study is justified by the diversity of situations represented in the thematic maps of distribution of essential activities described in Sections 3.1.1 and 3.1.2.

Despite the fact that the Eixample is considered the downtown of Barcelona [38], there are remarkable differences in the distribution of activities and the intensity with which they are carried out in the district. The analysis of essential urban services stresses internal nuances in their distribution and proportion and confirms this heterogeneity. By comparison with the centers of the old villages such as Gràcia, Sant Andreu or Sants, the isotropic nature of Cerdà's grid gives no clue as to which are the most central spaces within the grid. Nevertheless, the administrative delimitation establishes a total of six neighborhoods inside the district, from southeast to northwest, clockwise: Sant Antoni, Nova Esquerra de l'Eixample, Antiga Esquerra de l'Eixample, Dreta de l'Eixample, Sagrada Família and Fort Pienc (Figure 8 shows these administrative perimeters and some references to the main axes mentioned in following paragraphs).

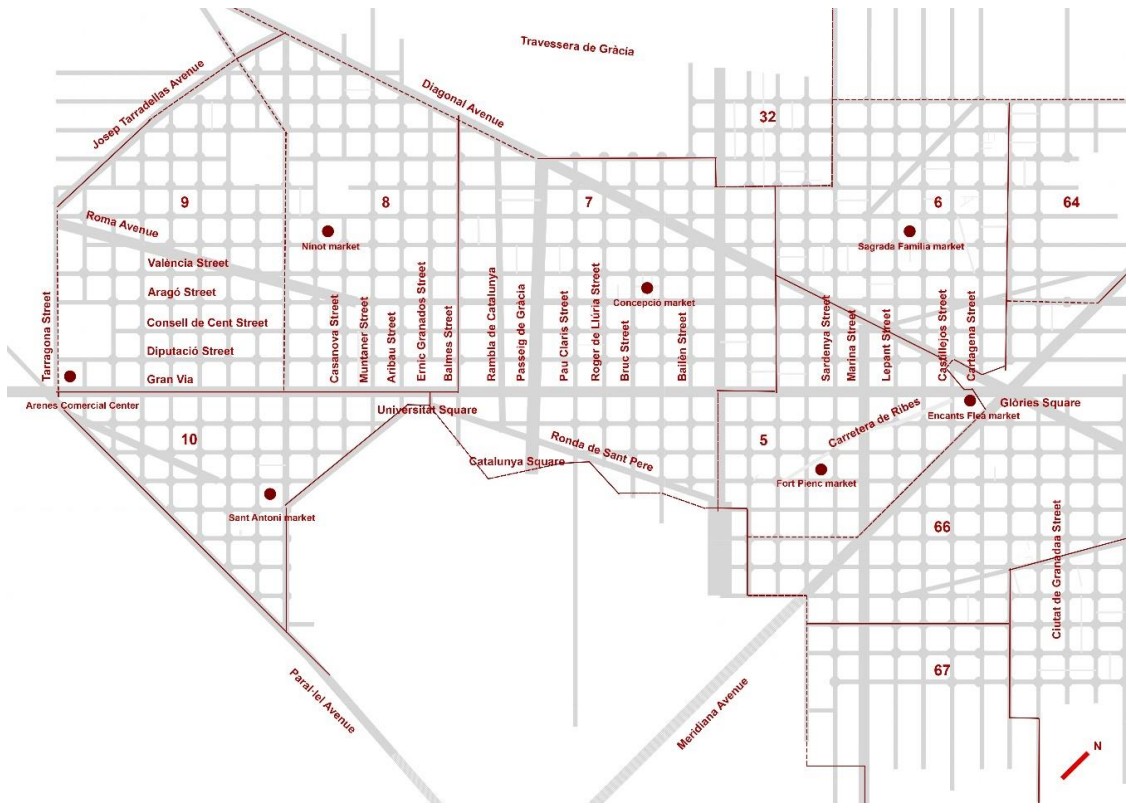

**Figure 8.** Eixample street layout and nomenclature of the main axis mentioned in the text.

Among the 14,006 active establishments in the Eixample, 3854 (27%) are located in La Dreta de L'Eixample (212.3 hectares), which is the administrative unit with a higher commercial supply index (CSI), set at 8.72 establishments per 100 inhabitants [28]. In this case, there is no doubt about the role of centrality of this area where the strong and diverse commercial offer coincides with a lower density of population together with a higher concentration of tertiary uses. On the other side of Balmes street, the Antiga Esquerra de l'Eixample (123.4 hectares) is the second most endowed neighborhood with 2730 establishments (19% of the city total) and a CSI of 6.33. The remaining four neighborhoods vary their relative position on this commerce endowment list: La Nova Esquerra (2163 establishments and a ratio of 3.69), La Sagrada Família (2114 and 4.07), Sant Antoni (1943 and 5.04) and Fort Pienc (1202 and 3.68).

Commercial performance decreases notably in the Poblenou area, where Cerdà's grid extends beyond the northern border of the Eixample district itself. The contiguous neighborhood of El Parc i la Llacuna del Poblenou only has 721 establishments but has still a CSI index higher than Fort Pienc, with 4.59 establishments per 100 inhabitants due to the lower residential density.

Besides the administrative position drawn from the municipal report of commercial activities of 2019, a more in-depth analysis of the grid data and spatial configuration reveals striking nuances within each neighborhood. This study is based on ongoing research by the authors [39] that analyzes the various proportion of uses on the ground floor of the Eixample in an area that includes several of the administrative districts. It encompasses the blocks between Tarragona street, Paral·lel, Josep Tarradellas and Diagonal, avenues Travessera de Gràcia, Ciutat de Granada street and the seafront to determine some factors in the mixedness that characterizes this area, which is usually considered as a model of the compact city.

The investigation starts by comparing an average block of the Eixample in the studied area, which has a total surface of 12,319 m$^2$ and, in normal conditions, an average of almost 37 active premises on the ground floor occupying 8172 m$^2$. Of these, the average commercial floor area is 6390 m$^2$ with 28.9 activities. The rest of the non-residential floor area is intended for an extended list of activities that can be grouped as hospitality, services, public facilities and production and logistics. A remaining set corresponds to undefined activities and closed establishments.

Three figures accurately describe the composition of the activities: two graphs focusing on blocks that have a density of activities as a reference for the quantification; and a drawing focused on the streets that evaluates the intensity of essential services as a reference for the qualification of public space. All graphs count the number of activities at ground floor level, excluding for the purpose of this research residential spaces, public facilities, production and logistics.

- Figure 9 shows the variety in the mixture. The ideal average block of this area presents a proportion of the number of activities divided into 26.6% of commerce (red), 28.2% of services (magenta), 10.9% of food trade (orange), 24.6% of hospitality (yellow) and 9.7% of facilities (blue). In general terms and with specific exceptions, all the blocks have 25% of the activities on the ground floor for services. The blocks between Balmes and Pau Claris streets, a large portion of Aribau street and some fragments of Consell de Cent street (on the left side), Gran Via (in the centre) and València street (both on the left and the right) are significant for having a much bigger proportion of activities aimed at commerce.

In an analysis of regular urban conditions, Balmes street (administrative limit of the neighborhood) also marks a limit to the left in which the number of activities is greater than in the rest of the district. Towards the northeast, Aragó and Lepant streets enclose a square delimited by the seafront in which the number of activities (that determines the diameter of the graph) is drastically lower. Within each of these areas, four clusters stand out, in which blocks with the greatest number of activities are concentrated (around three of the Eixample food markets: Sant Antoni, Ninot and Sagrada Familia, following a grouped concentration pattern, and on both sides of the Rambla de Catalunya, following a linear concentration pattern). In addition, three blocks in peripheral positions stand out from those that surround them also due to their greater number of activities (in Diagonal Avenue, between Casanova and Muntaner streets; in Travessera de Gràcia, between Castillejos and Cartagena streets; in Ronda de Sant Pere, between Roger de Llúria and Bruc streets).

- Figure 10 describes the proportion of essential services (in black) in relation to all the activities at street level (hatched with grey). Only 9% of the non-residential ground floor surface in the Eixample responds to the essential urban services analyzed in this research, which means that on average, there are 25.3 (86%) of other activities per block. Four blocks stand out for having a higher proportion of essential services than other activities: these are the four market halls, Sant Antoni, Ninot, Concepció, Sagrada Familia and Fort Pienc. In addition, another 14 blocks scattered around show values well above the 14% average of essential services. On the opposite side, blocks that in regular conditions host shopping centers, department stores or are located in known commercial axes (such as Passeig de Gràcia or Rambla Catalunya) stand out in the image for having a practically insignificant proportion of open activities.

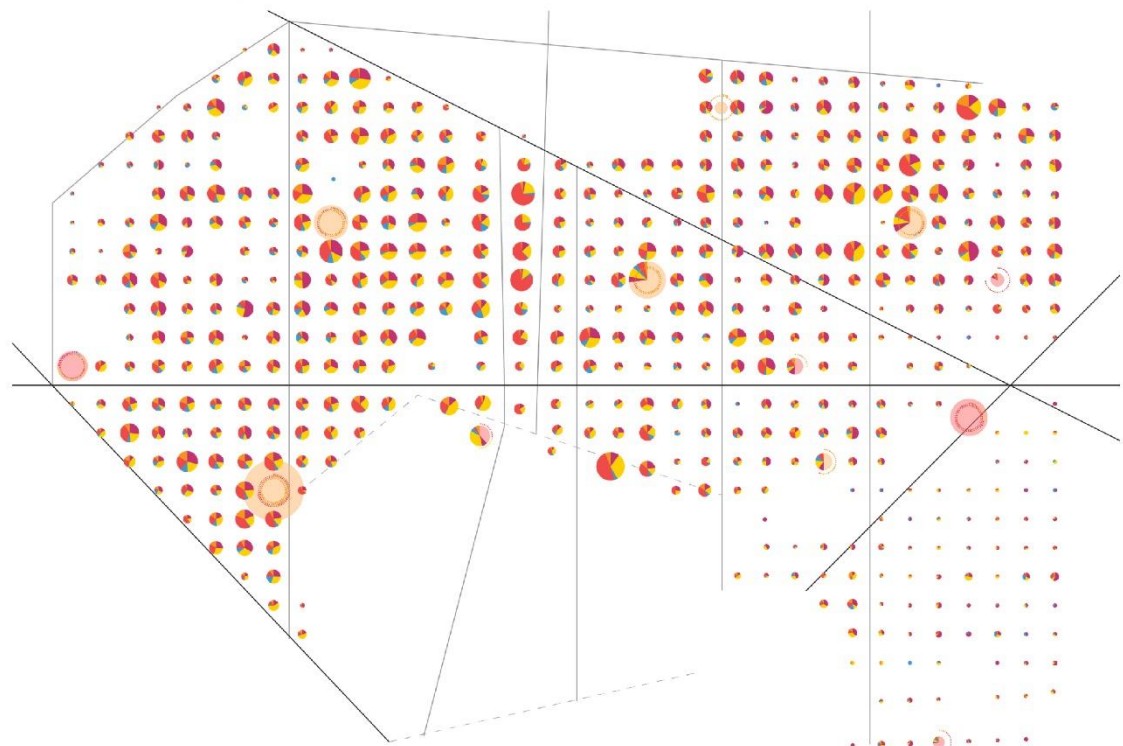

**Figure 9.** Activities mixture by blocks in the Eixample. Representation of the amount of activity units of each group through pie chart. Considerations: (1) the size of the circle depends on the total count of activities; (2) in markets and shopping centers (as considered by the Economic activities census), the activity count is made in perimeter dots; (3) if a pie chart is overlaid, it only represents activities of the block that are not part of the market or shopping center; and (4) blocks containing only public facilities are not shown. The main city axes overlapped.

- Figure 11 illustrates the real impact of essential activities on the streets: it compares continuous fronts with at least one essential commercial establishment (represented in red; blue shows those fronts with other activities or stores). This figure is clearly influenced by the lockdown situation in which the investigation was carried out, since the facade fragments represented in red were those with some commercial activity in operation, and those represented in blue, those that did not have any raised blinds.

The representation of the grid through essential services eliminates in most blocks one or two streets out of the four that define the emblematic octagon and its chamfers, blurring the characteristic, robust continuity of the Cerdà grid. The drawing shows three streets that are structured in urban terms but that nevertheless remain off from the perspective of essential services: Passeig de Gràcia, due to the global nature of the activities; Gran Via from Plaça Universitat to Plaça de les Glòries, with a minimum activity on the south façade; and Marina street. In addition, other streets like Enric Granados, Bailén or Sardenya (vertical) or the perpendicular Diputació or València (between Josep Tarrradellas and Diagonal avenues) disappeared from the grid.

On the left side of the Eixample, the intersection between Gran Via and Urgell street outlines two quadrants that behave differently: while the dense commercial network of Sant Antoni market keeps certain activities open in each street, Roma Avenue creates a void emphasized by the irregularity of the intersections. On the right side of the Eixample, Carretera de Ribes defines a kind of border of the activity of the center and signals the major inactivity of the streets in the area of Poblenou, where the regularity and continuity of grid activity completely disappears in the image.

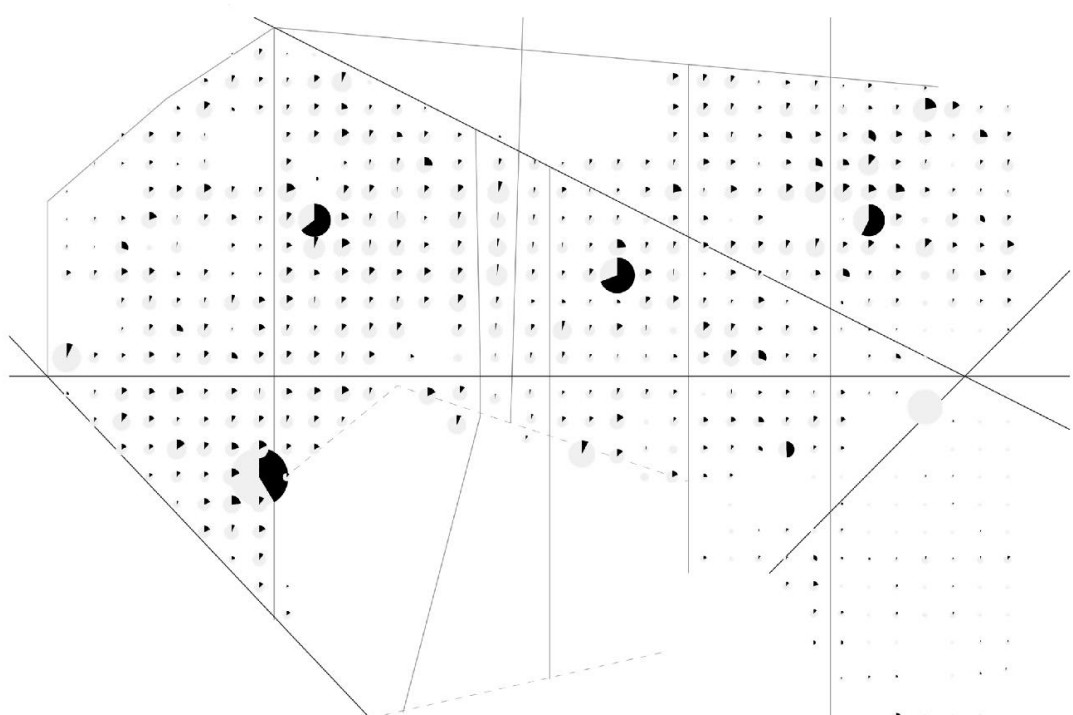

**Figure 10.** Proportion of essential services in relation to the total number of activities in the Eixample. Representation of the amount of activity units of each group through a pie chart. Considerations: (1) the size of the circle depends on the total count of activities; (2) black represents the amount of units that are open, grey the amount of activities that are closed; and (3) blocks containing only public facilities are not shown. The main city axes overlapped.

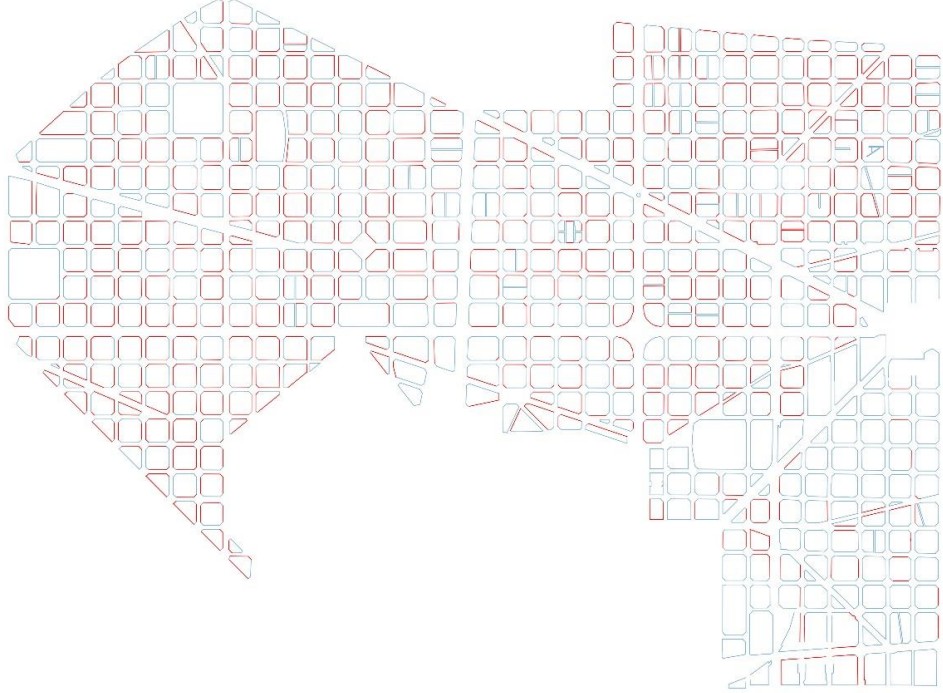

**Figure 11.** Streets with essential services in the Eixample (**red**) versus streets with other type of activities or without activities (**blue**).

## 4. Discussion and Conclusions

The analysis of the essential services in Barcelona offers new images that depict in detail the distribution of these activities in relation to the needs and concentration of the population. Some of the quantifications will soon be invalid, because the average life of a business in Barcelona is 9.6 years [28] and the economic crisis resulting from the Covid-19 lockdown is already changing the urban commercial landscape and will make the databases used in this research outdated. However, the physical space is still there and could be the scene for new approaches.

Through a holistic analysis of the location of the most basic services and commercial premises, differences in the essential urban mixture over the city can be measured accurately. At the same time, the functional characterization of different neighborhoods can be understood, including residential specialization or tertiary and tourism attractiveness. Nevertheless, a number of limitations conditioned the final results, including:

- The starting point of this research sets essential services as exclusive dependents on the resident population, but a number of floating people distorted the perfection of the model (among them, the workers in these essential services).
- The census of premises in Barcelona [28] lists 16,393 establishments without information, which represent 20% of the total number and some of which would have been very exceptionally open.
- Activities are measured in number but not in surface area: according to the conditions of essentiality defined in this paper, this does not seem as relevant, but the data could be used in further research.

Barcelona is a dense and compact city in its physical form but it is also intense in its activities, factors that imply proximity in essential services. Previous research from the perspective of proximity services and micro-journey analyses showed that there are no substantial variations between neighborhoods in Barcelona due to the consolidated and widely spread networked services [40]. Despite the fact that the outcome of the research confirms this general hypothesis, it additionally provides detailed data on nuances among parts of the city.

The analysis of physical proximity between services and their intensity of use through GIS mapping provides new insights into the neighborhood characterization. Equations combining the number of residents and the distribution of essential establishments depict alternative images to those that consider all the commerce in the city. The research reveals data from different aspects:

- In terms of quantification, almost 80% of the city's built-up plots have four of what are considered the essential services (pharmacies, grocery stores, supermarkets and bakeries) at a distance of less than a five minute walk. This figure also considers facilities, industrial buildings and infrastructure services. The percentage would be even higher if it were restricted to the urban footprint that houses residences and services. In addition, it can be said that the service is abundant since the presence of essential services in each proximity circle is numerous: each plot in the city has, on average, 75 essential establishments in a 400 m radius. These are outstanding figures considering that Barcelona is a city of more than 1.5 M inhabitants spread over an extension of almost 100 km$^2$.
- In terms of distribution throughout the city, one of the most interesting findings is that there is not only a center–periphery characterization in the traditional concentration of services in the city. More diverse situations exist: Nou Barris, a working-class residential neighborhood has similar figures and averages to central spots in the Eixample, which is a priori more dedicated to services; whereas the situation in Sarrià, an upper-middle class residential neighborhood in the limit of the municipality, is comparable to that in Poblesec, which is also residential, but close to the old town core. The focused analysis on the Eixample is particularly striking as it highlights internal dissimilarities within the homogenous grid.
- Finally, in terms of urban structure logics, a singularity of Barcelona should be highlighted in relation to basic services. Besides the relative isotropy of the essential services combined with

notable differences among neighborhoods, it is remarkable how the distribution of market halls, whether viewed as an aggregation of stalls or as buildings of a single activity, stands out in the image of the city since they agglutinate essential commercial activity around them. This means that the strong historical roots of their foundation are still valid due to the agglomeration force created by these urban poles.

This article presents a description of the methodology used in the research. It can easily be extrapolated to other cities to open up the discussion about the essentiality of services and the minimum distances to them. Beyond the overall results, the investigation reveals a methodological contribution that could be used to evaluate urban environments. The approximations that relate essential services with proximity and density described in Sections 3.1.1 and 3.1.2 could exploit the extreme values obtained in depth to determine four types of exceptional urban places:

- superplaces, where a high concentration of services coincides with high densities of the population using them;
- non-everyday neighborhoods, in which a high concentration of services responds to low rates of inhabitants so they could absorb, in terms of daily use, additional population without collapsing;
- inactive areas, with a small amount of daily activities and a low demand in the few people concentrated in them; and
- daily deserts, places that are not very active and yet have high demand, where services cater to a large population but plots have few services within their reach.

In the case of Barcelona, it is clear that there are very few non-everyday neighborhoods and daily deserts, and inactive areas are residual in number. However, an extension of the classification to a greater number of categories is presented to be able to include other urban situations and further the comparative analysis between metropolises, as suggested at the beginning of this section.

This research provides some valid figures and data on the logics between urban density and service distribution that can be tested in other cities following and enriching this methodological approach. The sum of the evaluation cases might make a substantial contribution to measure suitable urban compactness and density; two of the major issues that after the pandemic situation will remain at the forefront of the urban planning debate.

**Author Contributions:** Conceptualization, methodology and investigation, Carles Crosas and Eulàlia Gómez-Escoda; data curation, Eulàlia Gómez-Escoda writing—original draft preparation, Carles Crosas and Eulàlia Gómez-Escoda; writing—review: Carles Crosas; editing, Eulàlia Gómez-Escoda; visualization, Carles Crosas supervision, Carles Crosas and Eulàlia Gómez-Escoda. All authors have read and agreed to the published version of the manuscript.

**Funding:** This research received no external funding.

**Acknowledgments:** The authors would like to thank Mikel Berra-Sandin, student of Architecture at ETSAB, for his energy, dedication, drawings and ability to understand their purpose. He collaborated in this research through the scholarship "Beca de colaboración en Departamentos Universitarios" [Collaboration scholarship in University Departments] sponsored by the Ministerio de Educación y Formación Profesional [Ministry of Education and Professional Training].

**Conflicts of Interest:** The authors declare no conflict of interest.

# Appendix A

**Table A1.** Detailed description of places referenced in Figures 3, 5 and 6.

| | Detailed Description of Highlighted Places | | | | | |
|---|---|---|---|---|---|---|
| | **Markets** | **Supermarkets** | **Grocery Stores** | **Bakeries** | **Healthcare Centers** | **Pharmacies** |
| Figure 3 | Distribution of essential services by categories (from left to right and top to bottom: market halls, bakeries, supermarkets, grocery stores, pharmacies and healthcare facilities). | | Neighborhoods: Raval, Barceloneta and Poble Sec. In addition, some axes are highlighted in some sections: Carrer Marina; Travessera de Gràcia; Pi i Margall and the Avinguda de la Mare de Déu de Montserrat. | The nuclei of the old municipalities and some street axes: Travessera de Dalt, Carrer Astúries, Marià Aguiló, Joaquim Costa, Carrer de Sants (especially near the Sants and Hostafrancs market halls), Carrer Sèquia Comtal-Rogent axis (in the vicinity of Clot market hall). | Travessera de Gràcia between Gran de Gràcia and Passeig de Sant Joan. In addition, two mainly tourist areas concentrate a large number of establishments of this type: the Avinguda de Gaudí axis, near the Sagrada Familia, and the Born neighborhood. | Both sides of the Diagonal between Plaça de Maria Cristina and Plaça de Mossèn Verdaguer, on a strip of approximately 1.5 km, to the north and 1 km to the south. There is a particularly intense cluster next to the Ronda de General Mitre, between the streets of Numància and Muntaner. | |
| Figure 5 | Places with the highest population allocation for each category of essential services (from left to right and top to bottom: market halls, bakeries, supermarkets, grocery stores, pharmacies and healthcare facilities). | The market halls that serve the most population according to this theoretical model are Sants, Les Corts, Ninot, Sant Antoni, Clot, Sagrada Familia, Estrella and Mercè. | Significant axes that are highlighted in the drawing: Ronda Badal; Tarragona-Numància; and Urgell street. | In general terms, those to which the largest number of assigned persons correspond are located to the north of Ronda del Mig-Travessera de Dalt-Ronda Guinardó and east of Independència street. There are also some dense spots on Marina Street between Diagonal and Meridiana avenues; on the trapeze between Roma avenue, Urgell Street, Gran Via and Tarragona Street and in Vila Olímpica, at the end of Bogatell avenue. | Around Sants-Eixample Esquerra; around Glòries, Sant Martí and western 22@; and the areas behind the hills of El Coll, El Carmel and Guinardó. | Districts of Les Corts, Sant Gervasi-Galvany, La Salut and, especially, Sarrià and Sant-Gervasi Bonanova. Districts of Roquetes, Torre Baró, Ciutat Meridiana, Trinitat Nova y Vella, Sant Andreu and Bon Pastor. |

**Table A1.** *Cont.*

| | | Markets | Supermarkets | Grocery Stores | Bakeries | Healthcare Centers | Pharmacies |
|---|---|---|---|---|---|---|---|
| | **Detailed Description of Highlighted Places** | | | | | | |
| Figure 6 | Urban fabric according to the proximity between essential services by category (from left to right and top to bottom: market halls, bakeries, supermarkets, grocery stores, pharmacies and healthcare facilities). Proximity between service units is represented by quintiles (the darker the hatch, the higher the concentration). | There are tiny overlaps either in very central areas (Santa Caterina and Sant Josep markets), or in the most peripheral (Carmel and Horta or Canyelles and Guineueta, among others). | | | | Such as Via Augusta-Bonanova; Pi Margall or Tarradellas Diagonal with Carles III. | |

**Table A2.** Detailed description of places referenced in Figure 7.

| | **Figure 7: Areas with the Fewest Number of Essential Services** | | | |
|---|---|---|---|---|
| As defined in text | Areas pointed out for their past or current industrial use (or its proximity to it). | In the southern area, the contour line at an altitude 60 m over the sea is a first clear limit. | The topographical band defined around lines 120–130 over the sea. | The line drops in the Guinardó district to the lower reference line 90 m over the sea. |
| Detailed description of places | Passeig de la Zona Franca and La Marina del Prat Vermell on the southern side; around Poblenou in the center; and the Bon Pastor and Montsolís industrial areas close to Besòs River. In this case, the overlap with the topography us representative of the tradition of the industrial buildings occupying the least healthy and humid areas, as it was the case of the Besòs Delta later completely urbanized, but still recognizable in this map. | Coincides with the Diagonal axis (from the southern entrance of Barcelona to the crossing with Sarrià Avenue); at this point, the virtual limit jumps upwards to the very representative milestone of Passeig de la Bonanova, defining a line that follows approximately the trace of Ganduixer street. | Is the reference limit following the traces of Bonanova and Passeig de Sant Gervasi. The same altitude limit is representative in Vallcarca neighborhood and its main avenue. | Following Mare de Déu de Montserrat avenue and Passeig de Maragall, encompassing the few neighborhoods in Barcelona that are exceptionally not facing the sea, where the sharp topographical scheme of Barcelona becomes more complex. |

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
