# Peer review of "Mapping Food and Health Premises in Barcelona. An Approach to Logics of Distribution and Proximity of Essential Urban Services"

_ijgi, doi:10.3390/ijgi9120746_

Round 1
Reviewer 1 Report
Major
- The research objectives are unclear. I do not clearly understand what the authors would like to tell through this manuscript.
- Just delineating the proximity of essential services seems not that helpful to provide a better understanding of the Lockdown situations caused by COVID-19 pandemic.
- The structure of this manuscript looks a bit strange. I do not understand why the authors made the sub-section, called Figures, Tables and Schemes.
- There is no methodological advance. Currently, many studies to measure the proximity use the network analysis, but this study simply drew a 400m radius.
Minor
- Cartographies -> maps (line 76) ; Cartography refers to the study of maps.
- Font sizes are different (line 136 – 140)
- Figure number should be assigned by the order. For example, Figure 2 came (Line 157) earlier than Figure 1 (Line 242).
Reviewer 2 Report
The research presented in the article under review, I believe, is a model for the study of urban geography, in terms of examining the conditions that have arisen from the COVID-19 pandemic.
The esteemed authors have cleverly exploited the open spatial data offered by different services of Barcelona, ​​wishing to approach cartographically the changing habits of the daily life of the inhabitants of the city, under the forced lockdown imposed by the pandemic, and I believe that to a large extent they have succeeded.
The article under review is scientifically structured, supported by a sufficient number of bibliographic references and an extensive description of the parameters they use in order to approach the issue.
They have made the assumptions that are necessarily required to deal with their research approach reliably, but I will note here that I would expect a little more explanation regarding the assumption of 400 meters as a threshold in the measurement of essential services supply ... (lines 122-125).
The maps they have produced show these parameters reliably and according to the rules of thematic cartography. And I think they respond to the mapping of not-so-ordinary phenomena with quite a lot of success.
In summary, I believe that the research presented here is a very good approach to the phenomenon that the authors decided to describe and it is worth publishing.
Best regards
Reviewer 3 Report
The paper considers the availability of essential services during the lockdown in Barcelona in Spring 2020. The analysis is based on two main data sources, a data set for the population per block level of the city, and a data set for the location of businesses. Availability is evaluated in terms of proximity of the population to the nearest essential services facility. The results explore the proximity of different parts of the city with respect to six categories of essential services.
Comments:
- The paper presents a thorough analysis based on these two data sources, but its presentation requires some prior knowledge of the geography of Barcelona, and at times it seems it is better suited to local residents or authorities rather than general audience. I would suggest that the authors reduce the dependency on prior geographical knowledge, or provide it.
- The analysis is not really about the Barcelona lockdown. It is about the proximity of essential services, and as such it is independent of any particular lockdowns. This is particularly true as no data sets created during the lockdown are used (see next point). The authors should make this clear, and potentially revise the title.
- The analysis is limited by the data sources used. There are several other data sets mentioned, such as an open mobile phone geolocation dataset, and an air quality dataset, which are specifically collected during the lockdown. It would be nice if these could be included in the analysis and help provide additional insight.
- The generalizability of this kind of research is a very most important aspect. For example, how would one repeat the same analysis for a different city? What are the methods used, what are the alternatives, what are pros and cons? This is in part answered in Section 2 and in Section 4, but they should be isolated from the context of Barcelona and presented in a way that would promote repeatability in other cities.
Round 2
Reviewer 1 Report
Thank you for your efforts on the revision.
I am still not sure about the implications about what the authors would like to tell thorugh this manuscript.
Author Response
The authors would like to thank again Reviewer 1 for his/her efforts on the paper revision. Very diverse input from anonymous reviewers and suggestions from the special issue main editor have clarified notably the narrative of the paper and make clearer its contributions.
Reviewer 3 Report
I am happy with the changes implemented.
I would only advise that the legibility of Tables 1 and 2, which are inserted as bitmap images, is increased. Vector graphics could be used, or the resolution and compression could be improved.
Author Response
Tables 1 and 2 have been modified and inserted as tables instead of images.